# Associated risk factors of severe dengue in Reunion Island: A prospective cohort study

**Mathys Carras[1], Olivier Maillard[2,3]\*, Julien Cousty[1], Patrick Gérardin[2,3], Malik Boukerrou[4], Loïc Raffray[5,6], Patrick Mavingui[6], Patrice Poubeau[7], André Cabie[8,9], Antoine Bertolotti[3,7]**

**1** Intensive Care Unit, CHU Réunion, Saint Pierre, Reunion, France, **2** Department of Public Health and Research, CHU Réunion, Saint-Pierre, Reunion, France, **3** Clinical Investigation Center, INSERM CIC1410, CHU Réunion, Saint Pierre, Reunion, France, **4** Department of Gynecology and Obstetrics, CHU Réunion, Saint Pierre, Reunion, France, **5** Department of Internal Medicine, CHU Réunion, Saint Denis, Reunion, France, **6** UMR Processus Infectieux en Milieu Insulaire Tropical (PIMIT), CNRS 9192, INSERM 1187, IRD 249, Université de La Réunion, Sainte-Clotilde, Reunion, France, **7** Department of Infectious Diseases, CHU Réunion, Saint-Pierre, Reunion, France, **8** Department of Infectious and Tropical Diseases, CHU Martinique, Fort-de-France, Martinique, France, **9** Clinical Investigation Center, INSERM CIC1424, CHU Martinique, Fort-de-France, Martinique, France

\* olmaillard@yahoo.fr

**Data Availability Statement:** All relevant data are within the manuscript and its Supporting Information files. The data underlying the results

## Abstract

### Background

Since 2018, a dengue epidemic has been raging annually in Reunion Island, which poses the major problem of its morbidity and mortality. However, there is no consensus in the literature on factors associated with severity of illness. The objective of this study was to identify the factors associated with the occurrence of severe dengue (SD) according to the criteria adopted in 2009 by the World Health Organization (WHO), during the 2019 epidemic.

### Methodology/Principal findings

A total of 163 patients with RT-PCR-confirmed dengue were included in a multicenter prospective cohort study in Reunion Island between January and June 2019. Of these, 37 (23%) were classified as SD, which involves presentation dominated by at least one organ failure, and 126 (77%) classified as non-SD (of which 90 (71%) had warning signs). Confusion, dehydration, and relative hypovolemia were significantly associated with SD in bivariate analysis (p < 0.05). The factors associated with SD in multivariate analysis were a time from first symptom to hospital consultation over 2 days (OR: 2.46, CI: 1.42–4.27), a history of cardiovascular disease (OR: 2.75, 95%CI: 1.57–4.80) and being of Western European origin (OR: 17.60, CI: 4.15–74).

### Conclusions/Significance

This study confirms that SD is a frequent cause of hospitalization during dengue epidemics in Reunion Island. It suggests that cardiovascular disease, Western European origin, and delay in diagnosis and management are risk factors associated with SD fever, and that restoration of blood volume and correction of dehydration must be performed early to be effective.

presented in the study are available from the INSERM CIC 1410 (cic@chu-reunion.fr).

**Funding:** AC was supported by a grant from the French Ministry of Health (PHRC, 2009, n° 29-01), https://solidarites-sante.gouv.fr/systeme-de-sante-et-medico-social/innovation-etrecherche/ l-innovation-et-la-recherche-clinique/appels-a-projets/article/les-projetsretenus, and a grant from the French network for Research and Action targeting emerging infectious diseases (REACTING), https://aviesan.fr/aviesan/accueil/toute-lactualite/ reacting-une-approche-multidisciplinaire-pour-relever-le-defi-des-crisesepidemiques PM was supported by a grant of the European Regional Development Fund (ERDF) through the RUNDENG project (No. 20202640-0022937), http://www.reunioneurope.org/UE_beneficiaire_aides2014.asp The funders had no role in study design, data collection and analysis, decision to publish, or preparation of the manuscript.

**Competing interests:** The authors have declared that no competing interests exist

## Trial registration

NCT01099852; clinicaltrials.gov

## Author summary

Dengue fever is a viral disease transmitted by mosquitoes that threatens more than half of the world's population. This re-emerging disease predominates in tropical areas such as Reunion Island, but is also expanding in formerly temperate regions. Some patients with dengue may have early signs (warning signs) of life-threatening complications (dengue hemorrhagic fever/shock syndrome) and require hospitalization, but only a minority of patients will progress to severe dengue. Identification of patients at risk is crucial to deliver optimal treatment without saturating intensive care units during epidemic periods. To this end, we followed 163 hospitalized patients with dengue during the 2019 epidemic in Reunion Island. Nearly a quarter of the patients had a severe form of the disease. The presence of cardiovascular disease, Western European origin, and delay in diagnostic and management were the main risk factors. This observation underlines the importance of an efficient detection of vulnerable populations and of an early management based on rehydration to prevent the occurrence of severe dengue in Reunion Island.

## Introduction

Dengue is the most common mosquito-borne arbovirosis. In 2017, its incidence was estimated at more than 104 million people, of which 40,000 induced a death [1,2].

In 2019, in view of the re-emergence and rapid expansion of this arbovirosis, the World Health Organization (WHO) ranked dengue among the top ten global health threats [3]. Dengue is caused by four viral serotypes (DENV 1–4) with significant genotypic variability [4] and is characterized by a large clinical polymorphism with numerous signs of varying severity. In 2009, WHO proposed to distinguish between non-severe dengue, dengue with warning signs (WS) and severe dengue (SD), the latter being marked by major plasma leakage, severe bleeding and/or organ failure [5]. The case fatality can reach 20% in severe forms in the absence of early and appropriate medical management [6].

Reunion, a French island in the Indian Ocean with a population of 860,000, has been experiencing seasonal dengue epidemics since 2018. Nearly 70,978 confirmed cases, 2,672 hospitalizations, and 75 deaths have been reported up until May 2022 [7].

This re-emergence on the island has caused saturation of the healthcare system during epidemic peaks, necessitating a better estimation of each individual's risk of developing SD. The performance of prognostic factors for SD varies according to the studies and the definitions of organ failure used when defining SD [8]. The objective of this multicenter prospective cohort study, conducted in the university hospitals of Reunion Island, was to identify the clinical factors associated with the development of SD in our context.

## Methods

### Ethical approval

Ethical clearance was obtained by the French National Agency for the Safety of Medicines and Health Products (ANSM) (n°IDRCB 2010-A00282-37) and by the committee for the

protection of individuals (CPP Sud-Ouest and Outre-Mer III, 06/30/2010). Written and signed informed consent of all subjects was obtained and data was anonymized. This cohort study was reported according to the STROBE (Strengthening the reporting of observational studies in epidemiology) guideline (S2 Table).

## Study population

CARBO (Cohorte ARBOviroses) is a French multicentric prospective cohort study dedicated to the understanding of the acute stage of arboviral infections. It was proposed by the university hospital of Martinique and is registered on clinicaltrials.gov (NCT01099852). All patients suspected of dengue fever who were admitted to emergency departments of the University Hospital of Reunion from 1st of January to 30th of June 2019, were eligible. Patients were recruited if they had a positive blood sample for DENV by RT-PCR, and onset of symptoms ≤7 days (until 21 days for SD). They must have presented at least two symptoms among the following: fever (documented or declared), headache, rash, myalgia, arthralgia, abdominal pain or tenderness, bleedings and/or low platelets blood level below 150 G/l. Children younger than 6 years of age need to present fever (documented or declared by the family) with or without pain on an age-adapted visual analogue scale.

The Tropical Fever Core multiplex RT-PCR (Fast Track Diagnostics) was used for on-site diagnosis. It was a conventional two step real-time RT-PCR, which could detect dengue (without differentiation between serotypes), chikungunya and West Nile viruses, *Leptospira spp.*, *Rickettsia spp.* and *Salmonella spp.*, and *Plasmodium spp.*, which are pathogens that could circulate in the South West Indian Ocean region.

Serology was not routinely performed, but when there was a strong suspicion of dengue fever and a negative RT-PCR result. It was not systematically double checked three weeks later during the epidemic, or in ambulatory care, so the link between the data could not be made with confidence. For this reason, only cases with positive RT-PCR results were included in the study. Dengue reinfection was not systematically documented in 2019 because Reunionese people were considered naïve to dengue at this time, and serum neutralization was rarely performed.

## Data collection and follow-up

The primary outcome was the determinants associated with severity by comparing characteristics of SD patients with non-SD patients. SD was identified according to the WHO 2009 classification. However, the definitions of organ failure were those of the various learned societies concerned (S1 Table). Sociodemographic and clinical data were collected by a clinician at each physical exam of the patient via a standardized questionnaire at Day 0 (D0), D3, D5-7, D8-10, D21, D45, D90, D180. Ethnicity was identified by patient self-determination, and divided into six categories: existence of a parent or an ancestor from Sub-Saharan Africa, India, West Europe, South-east Asia; being Native American; or from other origin(s). Biological data were collected only in the acute phase from day 1 to day 10 of symptoms. All biological exams were carried out in the laboratories of the University Hospital of Reunion. All collected data were reported in a case report form and are detailed in Table 1.

## Data analyses

Quantitative variables were described with means and standard deviations while categorical variables were described as numbers and percentages.

A bivariate analysis was performed to compare clinical and laboratory features of DF patients according to severity of illness (defined by the WHO in 2009). Gaussian variables

**Table 1. Demographic, clinical and biological data in severe dengue and non-severe dengue, in DENV positive patients.** Reunion Island 2019 (N = 163).

| Variables | N | Non-severe dengue (%) n = 126 | Severe dengue (%) n = 37 | P-value |
|---|---|---|---|---|
| **Demographic data** | | | | |
| Male | 70 | 49 (39) | 21 (57) | 0.055[a] |
| Female | 93 | 77 (61) | 16 (43) | |
| Mean age (years), mean [StD] | 163 | 51.0 [20.7] | 53.1[22.4] | 0.596[d] |
| <18 years | | 6 (5) | 3 (8) | 0.464[a] |
| 18–65 years | | 81 (64) | 20 (54) | |
| >65 years | | 39 (31) | 14 (38) | |
| BMI (kg/m2), mean [StD] | 145 | 25.7 (5.0] | 25.8 [6.0] | 0.926[c] |
| BMI>30 | | 21 (17) | 5 (14) | 0.640[a] |
| Ethnic group | | | | |
| Africa | 163 | 4 (3) | 0 | 0.575[b] |
| India | 163 | 18 (14) | 3 (8) | 0.412[b] |
| Asia | 163 | 9 (7) | 4 (11) | 0.494[b] |
| Europe | 163 | 1 (1) | 6 (16) | 0.001[b] |
| America | 163 | 1 (1) | 0 | 1.000[b] |
| Others | 163 | 54 (43) | 16 (43) | 1.000[a] |
| **Healthcare circuit** | | | | |
| Delay from symptom to c(days), mean [StD] | 155 | 2.6 [2.3] | 3.8 [3.2] | 0.010[d] |
| Hospitalization | 163 | 91 (72) | 32 (87) | 0.063[a] |
| Length of stay (days), mean [StD] | 121 | 4.9 [2.6] | 7.1 [4.5] | 0.013[d] |
| Intensive Care Unit | | 19 (15) | 15 (41) | 0.002[a] |
| Medicine (Short/Medium stay unit) | | 80 (64) | 28 (76) | 0.235[a] |
| **Medical history and treatments** | 163 | | | |
| Hypertension | | 40 (32) | 15 (41) | 0.340[a] |
| Diabetes | | 27 (21) | 11 (30) | 0.314[a] |
| Dyslipidemia | | 10 (8) | 6 (16) | 0.204[b] |
| Cardiovascular disease | | 53 (42) | 23 (62) | 0.031[a] |
| Chronic kidney disease | | 14 (11) | 0 | 0.041[b] |
| Dialysis | | 4 (3) | 0 | 0.575[b] |
| Cirrhosis | | 0 | 0 | - |
| Peptic ulcer | | 5 (4) | 0 | 0.590[b] |
| Preventive anticoagulant | | 1 (1) | 1 (3) | 0.406[b] |
| Curative anticoagulant | | 8 (6) | 2 (5) | 1.000[b] |
| Antiplatelet agent | | 15 (12) | 5 (14) | 0.781[b] |
| **Clinical signs in the acute phase** | | | | |
| Temperature (°C), mean [StD] | 144 | 36.6 [0.8] | 36.6 [0.8] | 0.690[d] |
| Heart rate (bpm), mean [StD] | 143 | 77 [21.0] | 82 [18.0] | 0.179[d] |
| Mean blood pressure (mmHg), mean [StD] | 147 | 87 [16.0] | 83 [16.0] | 0.203[c] |
| Skin recoloration time >3 seconds | 163 | 6 (5) | 4 (11) | 0.237[b] |
| Oliguria | 163 | 2 (2) | 1 (3) | 0.543[b] |
| Coinfection | 163 | 12 (10) | 7 (19) | 0.146[b] |
| Thrombocytopenia <150 G/L | | 52 (41) | 19 (51) | 0.278[a] |
| Faintness | 163 | 17 (14) | 10 (27) | 0.063[a] |
| Confusion | 163 | 12 (10) | 12 (32) | 0.001[a] |
| **Warning signs** | 163 | | | |
| Warning signs (WS) | | 90 (71) | 30 (81) | 0.229[a] |

*(Continued)*

**Table 1.**  (Continued)

| Variables | N | Non-severe dengue (%) n = 126 | Severe dengue (%) n = 37 | P-value |
|---|---|---|---|---|
| Number of WS, mean [StD] | | 1.1 [1.1] | 1.5 [1.0] | 0.090[d] |
| Abdominal pain | | 55 (44) | 14 (38) | 0.528[a] |
| Persistent vomiting | | 15 (12) | 7 (19) | 0.281[b] |
| Clinical fluid accumulation | | 1 (1) | 0 | 1.000[b] |
| Mucosal bleed | | 35 (28) | 14 (38) | 0.248[a] |
| Lethargy, restlessness | | 5 (4) | 6 (16) | 0.018[b] |
| Liver enlargement > 2cm | | 2 (2) | 1 (3) | 0.541[b] |
| High hematocrit with low platelet count | | 14 (11) | 9 (24) | 0.054[a] |
| **Severity** | | | | |
| SOFA Score, mean [StD] | 26 | 4.4 [2.1] | 4.3 [3.0] | 0.981[d] |
| Shock | 163 | 0 | 14 (38) | <0.001[b] |
| Respiratory failure | 163 | 0 | 6 (16) | <0.001[b] |
| Severe bleeding | 163 | 0 | 14 (38) | <0.001[b] |
| Liver failure | 163 | 0 | 6 (16) | <0.001[b] |
| Kidney failure | 163 | 0 | 9 (24) | <0.001[b] |
| Heart failure | 163 | 0 | 0 | - |
| Neurologic failure | 163 | 0 | 2 (5) | 0.050[b] |
| Disseminated intravascular coagulation | 163 | 0 | 0 | - |
| Death | 163 | 0 | 1 (3) | 0.050[b] |
| Dehydration† | 163 | 73 (58) | 30 (81) | 0.009[a] |
| Hypovolemia‡ | 163 | 2 (2) | 5 (14) | 0.007[b] |
| **Biology** | | | | |
| Dengue virus reinfection | 163 | 8 (6) | 1 (3) | 0.685[b] |
| Sodium (mmol/L), mean [StD] | 136 | 136.8 [4.3] | 136.1 [4.8] | 0.454[d] |
| Haemoglobin (g/dL), mean [StD] | 138 | 14.1 [9.1] | 13.7 [2.7] | 0.763[d] |
| Hematocrit (%), mean [StD] | 138 | 38.9 [4.9] | 39.9 [7.4] | 0.497[d] |
| Leukocyte (G/L), mean [StD] | 138 | 5.6 [3.3] | 6.6 [4.8] | 0.222[d] |
| Neutrophils (G/L), mean [StD] | 133 | 4.1 [3.1] | 4.8 [4.7] | 0.337[d] |
| Lymphocyte (G/L), mean [StD] | 133 | 1.3 [4.1] | 1.0 [0.7] | 0.683[d] |
| Platelets (G/L), mean [StD] | 137 | 158 [81] | 173 [153] | 0.586[d] |
| Prothrombin Time (%),mean [StD] | 53 | 92 [22] | 82 [33] | 0.199[d] |
| ACT ratio, mean [StD] | 52 | 1.17 [0.22] | 1.44 [0.83] | 0.228[d] |
| Fibrinogen (g/L), mean [StD] | 47 | 3.9 [1.0] | 4.0 [1.5] | 0.677[d] |
| Albumin (g/L), mean [StD] | 8 | 44.0 [3.0] | 44.5 [2.1] | 0.848[d] |
| Protein (g/L), mean [StD] | 118 | 73.8 [6.7] | 72.9 [8.5] | 0.568[c] |
| Creatinine (μmol/L), mean [StD] | 135 | 108 [106] | 97 [57] | 0.550[d] |
| Urea (mmol/L), mean [StD] | 135 | 5.9 [4.0] | 6.2 [6.0] | 0.683[d] |
| AST (UI/L), mean [StD] | 131 | 73 [91] | 368 [746] | 0.036[d] |
| ALT (UI/L), mean [StD] | 131 | 50 [64] | 265 [650] | 0.076[d] |
| Total bilirubin (μmol/L), mean [StD] | 102 | 10.5 [13.7] | 9.9 [9.2] | 0.837[d] |
| Conjugated bilirubin (μmol/L), mean [StD] | 92 | 5.1 [10.9] | 5.4 [7.9] | 0.916[d] |
| Lipase (UI/L), mean [StD] | 97 | 51 [49] | 43 [27] | 0.435[d] |
| C-Reactive Protein (mg/mL), mean [StD] | 129 | 22 [38] | 22 [25] | 0.940[d] |
| Lactate (mmol/L), mean [StD] | 5 | 3.00 | 2.9 [1.4] | 0.977[d] |
| Troponin (ng/mL), mean [StD] | 8 | 0.019 [0.040] | 0.046 [0.051] | 0.123[d] |

(*Continued*)

**Table 1.** (Continued)

| Variables | N | Non-severe dengue (%) n = 126 | Severe dengue (%) n = 37 | P-value |
|---|---|---|---|---|
| Brain Natriuretic Peptide (pg/mL), mean [StD] | 5 | 512 | 4330 [2909] | 0.297[d] |

BMI: body mass index, WS: warning signs, HCT: hematocrit, SOFA: Sequential Organ Failure Assessment, DENV: dengue-virus, ACT: activated clotting time, AST: aspartate-amino-transferase, ALT: alanine-amino-transferase, StD: standard deviation

†Rehydration (oral or intravenous)

‡ Fluid resuscitation

P-values were calculated using

[a] Chi-square test

[b] Fisher exact test

[c] Student t test

[d] Wilcoxon Mann Whitney test

were compared using the Student's t-test, while non parametric continuous variables were compared using the Wilcoxon Mann Whitney test. The normality of the distributions was checked by the Shapiro-Wilk test, and the homogeneity of variances by the Levene test. Categorical variables were compared using Chi-square test or the Fisher's exact test, as appropriate.

A multivariate logistic regression was performed to identify factors associated with SD. Variables with a $p$-value <0.2 in bivariate analysis were entered into a multivariate model. In the case of collinearity, detected by using the variance inflation factor (VIF), the most relevant variable was retained in the model. A backward stepwise automated procedure was used to select factors associated with $p$-value <0.05 in the final model. Adjusted odds ratios (OR) and their 95% confidence intervals (95%CI) were derived from the regression coefficients. Data were analyzed using SPSS software (IBM SPSS 23.0, IBM Corp. Armonk, NY, USA). All tests were two-tailed and the significance level was set at 0.05.

## Results

### Characteristics of the study population

In total, of 172 eligible patients who consented to participate, 163 were included in the study (Fig 1). The average age was 52 years old, and there were more women (57%) than men. Indian origin was the most represented (13%). Cardiovascular comorbidities were the most reported medical background (46%), followed by hypertension (34%), diabetes mellitus (23%), and obesity (16%) (Table 1). The most common serotype identified in 2019 was DENV-2.

The incidence of clinical and biological signs and their chronology are detailed in Fig 2. The clinical signs most frequently recorded in the acute phase were fever (93%), asthenia (94%), anorexia (83%), headache (59%), musculoskeletal signs with myalgia (69%), arthralgia (59%) and spinal pain (50%). Thrombocytopenia was present in 44% of patients. The earliest clinical signs of dengue were, in chronological order: vomiting (median 2 days), lipothymia (median 2.5 days), fever (median 3 days), and diarrhea (median 3 days). The maximum duration of symptoms was 150 days for asthenia and 167 days for arthralgia (Fig 2).

Of the 163 patients included, 37 (23%) were classed as severe of whom 24 (65%) at the time of recruitment, 14 (38%) with shock, 14 (38%) with severe hemorrhage, 23 (62%) with severe organ failure dominated by kidney failure (n = 9) and 1 (3%) which resulted in death. Hospitalization was required in 91 (72%) non-severe cases versus 32 (87%) severe cases (p = 0.063). Coinfections were not associated with severity (p = 0.146) and out of 19 (12%) co-infected patients, 9 urinary tract infections, 5 bacteremias, 4 pulmonary infections, 3 gastroenteritis

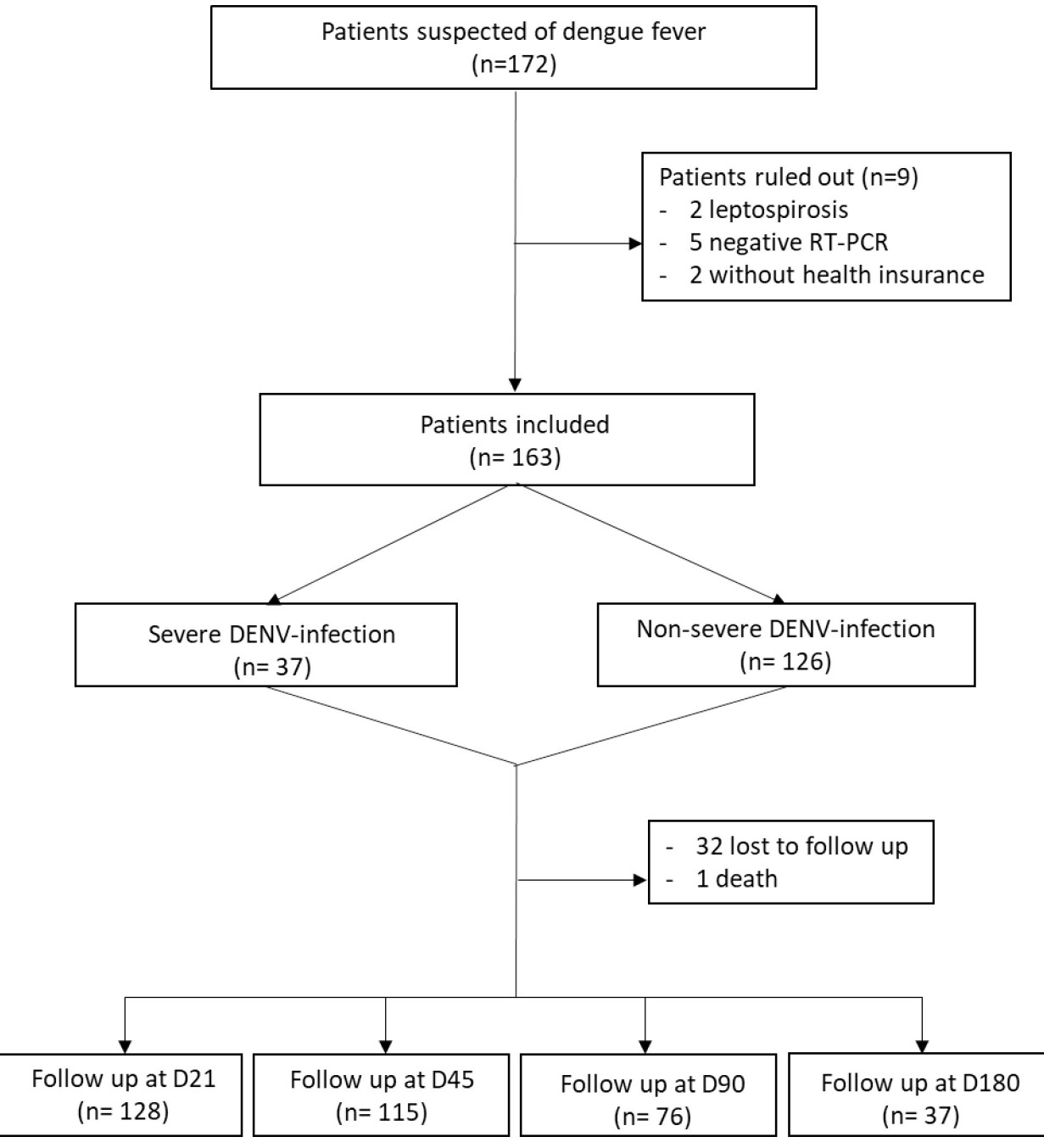

**Fig 1. Study population, CARBO cohort study, Reunion Island 2019.** Legend: DENV: dengue virus, RT-PCR: reverse transcriptase polymerase chain reaction, D: day.

and 1 skin infection were reported. They were present at admission in 15 (79%) patients. In bivariate analysis, the main factors associated with SD were: Western European origin (p = 0.001), a history of cardiovascular disease (p = 0.031), confusion (p = 0.001), dehydration defined as the need for oral or intravenous rehydration (p = 0.009), relative hypovolemia defined as the need for at least one vascular filling (p = 0.007), an increase in aspartate-amino-transferase (ASAT) level (p = 0.036) and the delay from symptom to consultation (p = 0.010)

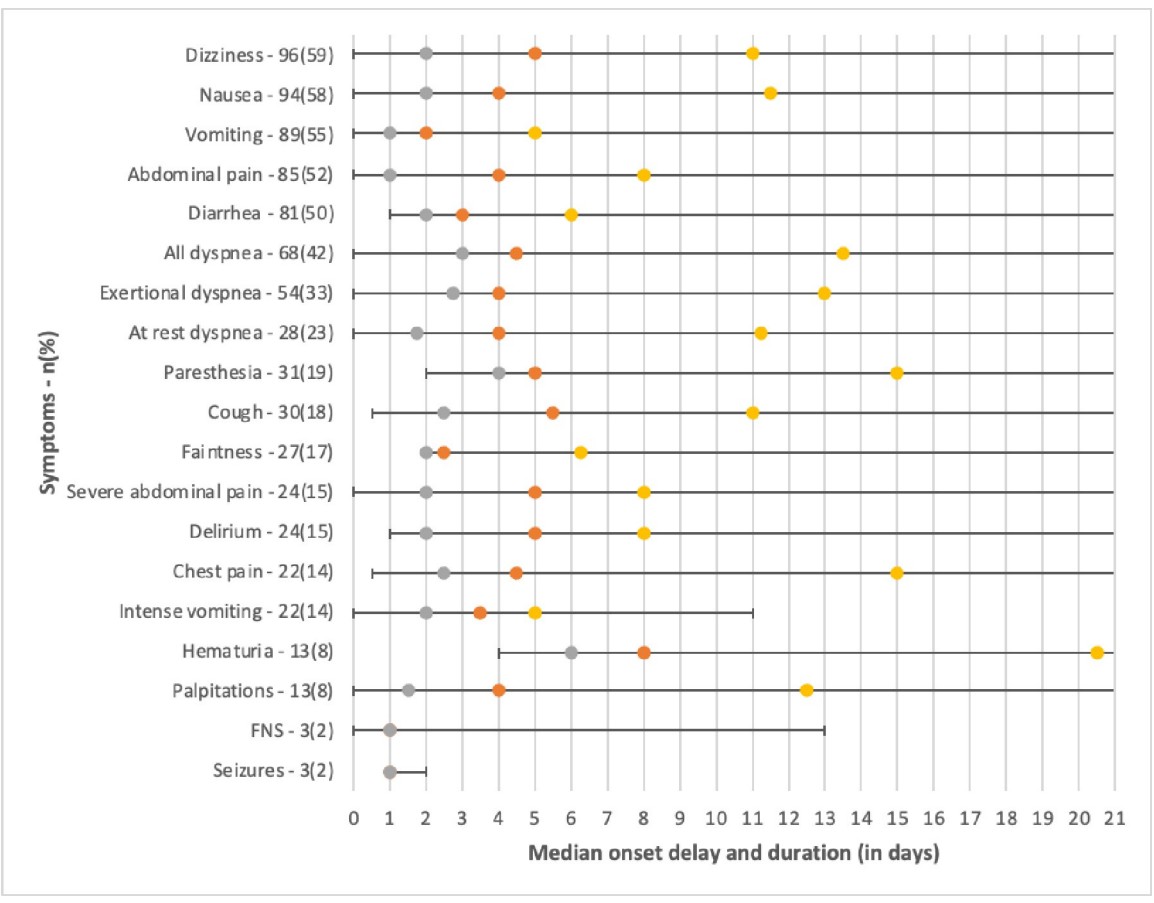

**Fig 2. Incidence and timing (time to illness onset and duration) of dengue manifestations, in patients with positive Dengue virus RT-PCR. CARBO cohort, Reunion Island 2019.** Legend: Grey: first quartile, orange: median, yellow: third quartile.

(Table 1). The mean length of stay was longer in cases of SD than in cases of non-SD (p = 0.013).

Multivariate analysis identified an increased risk of SD in patients with a time from first symptom to consultation over 2 days (OR: 2.46, CI: 1.42–4.27), a history of cardiovascular disease (OR: 2.75, 95%CI: 1.57–4.80), being of Western European origin (OR: 17.60, CI: 4.15–74) (Table 2).

## Discussion

The prevalence of SD was 23% in this study. The warning signs had a low discriminatory value as their proportion was similar between the two groups. There was a co-circulation of two

**Table 2. Factors associated with severity of dengue in adults with positive RT-PCR at disease onset. Multivariate analysis using logistic regression. CARBO cohort study, Réunion Island 2019 (N = 163).**

| Variables | Adjusted Odds Ratio (95%CI) | P-value |
|---|---|---|
| Time elapsed between illness onset and presentation to hospital > 2 days | 2.46 (1.42–4.27) | 0.036 |
| West European origin | 17.60 (4.15–74) | 0.011 |
| Cardiovascular disease | 2.75 (1.57–4.80) | 0.021 |

95%CI: 95% confidence interval

serotypes, dominated by DENV-2 [9]. This result is similar to that of a study carried out in French Guyana which identified 16% of SD during an epidemic mixing the same two serotypes [10]. Serotype 2 is the most virulent strain according to the literature [11]. The co-circulation of several serotypes in Reunion Island raises the threat of more serious future epidemics linked to the ADE (antibody-dependent enhancement) phenomenon observed during secondary infections [12].

A history of cardiovascular disease was associated with SD according to a recent meta-analysis, which reported that hypertension and diabetes were predictive factors for SD [13]. These comorbidities lead to dysfunction of the capillary endothelium, with increased vascular permeability and the occurrence of hypovolemic shock responsible for SD. Moreover, the co-existence of these comorbidities also favors organ decompensation [14,15]. The population of Reunion Island is particularly exposed to cardiovascular risk factors, which are potentially linked to genetic factors [16–18].

Our work suggests that Western European origin is also a risk factor for progression to SD. These results are similar to those of a study conducted in the Cuban population. In that study, the authors suggested that polymorphism in genes associated with the immune response may be responsible for the increased susceptibility of European ethnicity to ADE [19]. Risk factors associated with SD, such as cardiovascular disease and Western European origin, raise concerns that the recent spread of dengue to these new areas may increase the incidence of SD and its lethality [20,21].

Dehydration defined as the need for oral or intravenous rehydration was associated with SD. Rehydration is the main treatment for dengue with warning signs to prevent progression to a severe form. It may cause confusion or lipothymia, which are two warning signs present in one third of SD cases in our study. These elements suggest that dehydration plays a predominant role in the critical phase, just as it can cause complications in other arboviroses [22–24]. A late consultation, which delays intravenous rehydration, could also explain the increased risk of SD.

Indeed, delay in diagnosis and management was also associated with SD. Severe bleedings were the only severity component to be significantly associated with a consultation time greater than 2 days after onset. Although the date of illness onset is declarative, this is consistent with the literature which reports that delay in diagnosis is associated with severity including severe bleedings, complications, and mortality of dengue [6,25–27].

No patient with chronic kidney disease presented with SD. The difficulty in applying the KDIGO definition could be the cause. Indeed, urine quantification was not routinely measured in conventional hospitalization, baseline creatinine was not always available, and isolated increase in blood creatinine was insufficient because of already high levels in patients with chronic kidney disease. We used Stage 3 of the KDIGO classification to define acute kidney failure. By choosing a severe and restrictive definition of kidney failure to target severe cases in an inpatient cohort, cases of moderate acute kidney failure were not selected. With a less restrictive definition, eighteen patients had acute kidney failure according to KDIGO Stage 2, including 3 patients with chronic kidney disease.

The strengths of this study were (i) the early inclusion and prospective follow-up of patients, (ii) systematic clinical examination by a clinical investigator at admission and up to day 10, and (iii) use of the definitions of each learned society to estimate organ failure. There were, however, limitations within this study: (i) it was a cohort not derived from a consecutive series, or randomly sampled among patients, (ii) the cohort represented 19% of patients who consulted the hospital with PCR-confirmed dengue in 2019 in Reunion Island according to epidemiological data [7] (these results were related to a time-consuming inclusion of patients in an epidemic context and a large number of patient refusals), (iii) the use of a uniform

definition of SD for all age groups could lack reproducibility in the pediatric population, especially for the shock criterion and the biological variables of liver and kidney failure, and (iv) the viral serotype of dengue patients could not be identified in all patients during the epidemic. A serum library was established, which will allow for future studies.

This first prospective study conducted during a dengue epidemic in Reunion Island allowed us to describe the risk factors specific to the Reunionese population and the evolution of the disease among patients who attended the hospital. Our study highlights the poor performance of warning signs in predicting SD, as well as the variability of the criteria for admission to intensive care depending on the clinician. Besides a history of cardiovascular disease and Western European origin that were associated with SD in our setting, the time to diagnosis and management of dengue is crucial, and rehydration appears to play a central role in preventing progression to SD.

Our results will be used in other large-scale studies during future dengue epidemics in Reunion Island, which will allow the identification of predictive factors in the evolution towards a more severe form of the disease, as well as the drafting of referral protocols to specialized services adapted to the profile of the patient.

## Supporting information

**S1 Table. Definitions of organ failure according to learned societies. Legend:** SBP: systolic blood pressure, DBP: diastolic blood pressure, SpO2: saturation pulse oxygen, ECG: electrocardiography, KDIGO: Kidney Disease Improving Global Outcomes, AST: aspartate-aminotransferase, ALT: alanine-amino-transferase
(DOCX)

**S2 Table. STROBE (Strengthening the reporting of observational studies in epidemiology) checklist for cohort studies.**
(DOCX)

## Acknowledgments

We would like to thank Drs E. Antok, Q. Balacheff, E. Barange, J.P. Becquart, L. Brochet, R. Chane Teng, R. Crouzet, A.B. Da Silva Gomez, C. Daubard, A. Desvergez, D. Khemiri, K. Larsen, M. Lemeur, L. Raffray, M. Ruin, F. Tixier, N. Ebran, B. Fontaine, C. Hebert, D. Hirschinger, E. Huchot, E, Jarlet, H. Flodrops, M. Lafon, A. Laval, O. Lamouret, J. Lemant, J.C. Maiza, R. Manaquin, R. Perrin, A. Plantier, C. Schweizer, L. Thibault. We also would like to thank V. Grondin, J. Jean-Marie, I. Calmont, J. Ruiz, and our copy editor Jennifer Sanders. The protocol was prepared with the help of the INSERM Research and Action Targeting Emerging Infectious Disease (REACTing) network.

## Author Contributions

**Conceptualization:** André Cabie.

**Data curation:** Antoine Bertolotti.

**Formal analysis:** Mathys Carras, Olivier Maillard, Antoine Bertolotti.

**Investigation:** Mathys Carras, Julien Cousty, Malik Boukerrou, Loïc Raffray, Patrice Poubeau, Antoine Bertolotti.

**Methodology:** Olivier Maillard, André Cabie, Antoine Bertolotti.

**Supervision:** Antoine Bertolotti.

**Validation:** Julien Cousty, Antoine Bertolotti.

**Visualization:** Patrick Gérardin, Malik Boukerrou, Loïc Raffray, Patrice Poubeau.

**Writing – original draft:** Mathys Carras, Julien Cousty, Antoine Bertolotti.

**Writing – review & editing:** Olivier Maillard, Patrick Gérardin, Loïc Raffray, Patrick Mavingui, Patrice Poubeau, André Cabie, Antoine Bertolotti.

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
