## [Decision Letter · Decision Letter 0]

9 Nov 2022

Dear Dr Maillard,

Thank you very much for submitting your manuscript "Associated risk factors of severe dengue in Reunion Island: A prospective cohort study" for consideration at PLOS Neglected Tropical Diseases. As with all papers reviewed by the journal, your manuscript was reviewed by members of the editorial board and by several independent reviewers. In light of the reviews (below this email), we would like to invite the resubmission of a significantly-revised version that takes into account the reviewers' comments. 

The manuscript is well written and covers an important topic that will be interesting to readers. As described in the detailed reviewer feedback, much more detail is needed in the methods section, including an in depth description of the univariate and multivariate analyses. Prior to resubmission the authors will need to review and respond to all reviewer comments. A more detailed description of analytical methods and results is expected as part of this revision.

 We cannot make any decision about publication until we have seen the revised manuscript and your response to the reviewers' comments. Your revised manuscript is also likely to be sent to reviewers for further evaluation.

Sincerely,

Mellisa Roskosky, Ph.D

Academic Editor

Elvina Viennet

Section Editor

The manuscript is well written and covers an important topic that will be interesting to readers. As described in the detailed reviewer feedback, much more detail is needed in the methods section, including an in depth description of the univariate and multivariate analyses. Prior to acceptance the authors will need to review and respond to all reviewer comments. A more detailed description of analytical methods and results is expected as part of this revision.

Reviewer's Responses to Questions

**Key Review Criteria Required for Acceptance?**

**Methods**

-Are the objectives of the study clearly articulated with a clear testable hypothesis stated?

-Is the study design appropriate to address the stated objectives?

-Is the population clearly described and appropriate for the hypothesis being tested?

-Is the sample size sufficient to ensure adequate power to address the hypothesis being tested?

-Were correct statistical analysis used to support conclusions?

-Are there concerns about ethical or regulatory requirements being met?

Reviewer #1: (No Response)

Reviewer #2: The objectives are clearly stated and hypothesis covered the factors included by the WHO classification. The design is adequate.

The population description requires more details for those who are not local to Reunion Island, the issue is what they cover in the classification of ethnic groups and how other groups are classified (maybe they include all in other?).

What is the definition of Indian and Asian ethnicities please specify (line 98 , page 10)

 The sample size is adequate for most analysis.This number of cases for those younger than 18 is relatively small. This can limit the interpretation. The description of the statistical analysis seems correct but lacks some details specially in the clear flow in the use of specific tests

No concerns about ethical or regulatory requirements.

Reviewer #3: There is a need to revamp method section. It is not clear on the type of study design and measure used to assess difference between severe and non-severe dengue. Study population is not clearly defined and sample size was not reported.

**Results**

-Does the analysis presented match the analysis plan?

-Are the results clearly and completely presented?

-Are the figures (Tables, Images) of sufficient quality for clarity?

Reviewer #1: (No Response)

Reviewer #2: The results seem according to the analysis plan however there are some issues with the interpretation of the multivariate analysis, the ORs and CI for the variable Age provide the idea of a protective effect in general, but affirming that population under 18 years is at high risk requires to show the analysis of the different age groups and ORs and CI greater than 1 (maybe they just need to do it more explicit in the manuscript). The multivariate analysis table should be in the main text just like the descriptive statistics

Reviewer #3: The main finding in placed in the supplementary file. It has to be relocated in the main content.

**Conclusions**

-Are the conclusions supported by the data presented?

-Are the limitations of analysis clearly described?

-Do the authors discuss how these data can be helpful to advance our understanding of the topic under study?

-Is public health relevance addressed?

Reviewer #1: (No Response)

Reviewer #2: Most of the conclusions are well supported by the data but identifying the age as risk factor requires at least to make explicit the work done with that variable and explaining how univariate analysis contributed/not contributed to reach the final model

Limitations are well described. I found a limited discussion of the contribution of this findings to the understanding of severe dengue and the most recent classification issued by WHO.

Public health relevance is addressed.

Reviewer #3: Conclusion is presented according to the finding of the study

**Editorial and Data Presentation Modifications?**

Reviewer #1: (No Response)

Reviewer #2: Provide more information about the breakdown of the population in ethnic groups, and clarify the process of identification of Age (and the category < 18 yrs as risk factor, State with clarity if the data support the serotype 2 as responsible for the described epidemic.

I think this modifications/additions qualify as minor revision

Reviewer #3: (No Response)

**Summary and General Comments**

Reviewer #1: In this cohort study, authors recruited confirmed dengue patients within 7 days of symptom onset and clinical and biological data was collected at various times post recruitment. Bivariate and multivariate analysis was performed to identify risk factors that are associated with developing severe dengue.

This may be the first study from Reunion Island but risk factors associated with severe dengue are extensively studied in the literature.

Please clarify in the text what symptom/s were used for patient recruitment.

What percentage of patients met the case definition of severe dengue at the time of recruitment?

Confusion is a subjective factor and therefore cannot be used for prediction. Authors seem to indicate in the abstract and conclusion that identified factors can predict who will develop severe dengue disease but in realty these are risk factors that are associated with severity and may or may not accurately predict severity of a disease. For example, diabetes, hypertension and various other factors correlated with severe dengue in various other dengue studies but did not associate in this cohort.

Line 108 authors meant p values of < 0.02 or 0.2? Age was not significantly associated in the bivariate analysis and so why was it included in the multivariate analysis?

Viral titers or ns1 levels will be important to report in this dataset.

Please describe which co-infections were reported.

Dengue reinfection- how it’s determined and whether it was homologous or heterologous reinfection? Please describe in the text.

Multivariate- what set of independent variables were used in multivariate analysis? Full table should be reported in the supplemental file.

For some factors such as age less than 18years and Western European the sample size is too small to make any meaningful determination.

Reviewer #2: My general impression is that this study was well designed and conducted considering the clinical setting and the unpredictable flow of cases in an epidemic, however the manuscript needs to be expanded to explain the process of handling the variables in the multivariate analysis compared with the univariate analysis

Reviewer #3: This study warrants major revision with special focus on method section

PLOS authors have the option to publish the peer review history of their article (what does this mean?). If published, this will include your full peer review and any attached files.

Reviewer #1: No

Reviewer #2: No

Reviewer #3: No
---

## [Decision Letter · Decision Letter 1]

16 Feb 2023

Dear Dr Maillard,

Thank you very much for submitting your manuscript "Associated risk factors of severe dengue in Reunion Island: A prospective cohort study" for consideration at PLOS Neglected Tropical Diseases. As with all papers reviewed by the journal, your manuscript was reviewed by members of the editorial board and by several independent reviewers. The reviewers appreciated the attention to an important topic. Based on the reviews, we are likely to accept this manuscript for publication, providing that you modify the manuscript according to the review recommendations. 

This version of the manuscript is much improved. There are a few minor revisions that will need to be addressed prior to publication. These revisions include adding some additional detail in the methods section that were provided in your response to reviewers of your original submission and expanding on your conclusions related to between disease onset and hospital presentation. Please see reviewer comments for additional details.

Sincerely,

Mellisa Roskosky, Ph.D

Academic Editor

Elvina Viennet

Section Editor

This version of the manuscript is much improved. There are a few minor revisions that will need to be addressed prior to publication. These revisions include adding some additional detail in the methods section that were provided in your response to reviewers of your original submission and expanding on your conclusions related to between disease onset and hospital presentation. Please see reviewer comments for additional details.

Reviewer's Responses to Questions

**Key Review Criteria Required for Acceptance?**

**Methods**

-Are the objectives of the study clearly articulated with a clear testable hypothesis stated?

-Is the study design appropriate to address the stated objectives?

-Is the population clearly described and appropriate for the hypothesis being tested?

-Is the sample size sufficient to ensure adequate power to address the hypothesis being tested?

-Were correct statistical analysis used to support conclusions?

-Are there concerns about ethical or regulatory requirements being met?

Reviewer #1: (No Response)

Reviewer #2: The objectives are clearly stated, and the study design in general is adequate to pursue the mentioned objectives. The description of the population is much clear than before and it contain the elements to test the hypothesis. The sample size goes according to the possibilities to gather a prospective cohort. Statistical analysis is fine and consistent to previous studies, and they support the conclusion. They followed a conventional method to reach their conclusion.

I don't find ethical or regulatory issues to be solved.

**Results**

-Does the analysis presented match the analysis plan?

-Are the results clearly and completely presented?

-Are the figures (Tables, Images) of sufficient quality for clarity?

Reviewer #1: (No Response)

Reviewer #2: Analysis was done according to the plan stated by the authors, results are completely presented, Tables reflect the analysis work done by the authors and they convey well the results they obtained. My only concern is how the presentation of results provide enough basis to discuss a richer discussion.

**Conclusions**

-Are the conclusions supported by the data presented?

-Are the limitations of analysis clearly described?

-Do the authors discuss how these data can be helpful to advance our understanding of the topic under study?

-Is public health relevance addressed?

Reviewer #1: (No Response)

Reviewer #2: The conclusions are supported by the data but they don't expand on the implication of one of the severe dengue predictors:

Time elapsed between illness onset and presentation to hospital. The authors should elaborate what is the practical meaning of that: failure in the early detection of cases, lack of information in the population. The other element to hypothesize is how the cardiovacular co-morbidities interact with that delay.

In general terms the limitations were mentioned . The authors provide the implications of their findings to understand the local characteristics of Severe dengue and how they are consistent to the general epidemiology of dengue.

Even when they approach from a clinical perspective they touch well the Public Health implications.

**Editorial and Data Presentation Modifications?**

Reviewer #1: (No Response)

Reviewer #2: My suggestion is to discuss the predictors and their potential interactions (based on other international references). Once they do that the paper is good to accept. I think this paper qualify for Minor revision.

**Summary and General Comments**

Reviewer #1: [Comment: Please describe which co-infections were reported. 

Response: Of 19 (12%) co-infected patients, 9 urinary tract infections, 5 bacteriemia, 5 4 pulmonary infections, 3 gastroenteritis and 1 cutaneous infection, were reported. Fifteen (79%) were diagnosed at admission. Five patients had complications related to their coinfection. As coinfections were not associated with severity, no more information was added in the manuscript.]

Comment: Please add above response information to the method section since related data is presented in table-1.

[Comment: Dengue reinfection- how it’s determined and whether it was homologous or heterologous reinfection? Please describe in the text. 

Response: Dengue reinfection was not systematically documented in 2019 because Reunionese people were considered naïve to dengue at this time, and serum neutralization was not routinely done.]

Comment: Please add in the method section that DENV re-infection was determined based on the presence of DENV reactive IgG at the time of enrollment using Dengue Duo kit or howsoever authors determined it, if differently.

Reviewer #2: The manuscript was improved and the suggestions were addressed by the authors. No additonal comments

PLOS authors have the option to publish the peer review history of their article (what does this mean?). If published, this will include your full peer review and any attached files.

Reviewer #1: No

Reviewer #2: Yes: Eduardo Fernandez C.

Figure Files:

Data Requirements:

Reproducibility:

References

---

## [Editor Report · Decision Letter 2]

21 Mar 2023

Dear Dr Maillard,

We are pleased to inform you that your manuscript 'Associated risk factors of severe dengue in Reunion Island: A prospective cohort study' has been provisionally accepted for publication in PLOS Neglected Tropical Diseases.

Best regards,

Mellisa Roskosky, Ph.D

Academic Editor

Elvina Viennet

Section Editor

Thank you for this thorough and thoughtful revision and response to comments.

---

## [Editor Report · Acceptance letter]

12 Apr 2023

Dear Dr Maillard,

We are delighted to inform you that your manuscript, "Associated risk factors of severe dengue in Reunion Island: A prospective cohort study," has been formally accepted for publication in PLOS Neglected Tropical Diseases.

Best regards,

Shaden Kamhawi

co-Editor-in-Chief

Paul Brindley

co-Editor-in-Chief
